# A New Formula for Vietnamese Text Readability Assessment

## Abstract

Text readability has an important role in text drafting and document selecting. Researches on the readability of the text have been made long ago for English and some common languages. There are few researches in Vietnamese text readability and most of them are performed from more than two decades ago on very small corpora. In this paper, we build a new and larger corpus and use it to create a newer formula to predict the difficulty of Vietnamese text. The experimental results show that the new formula can predict the readability of Vietnamese documents with over 80% accuracy.

## 1 Introduction

Text readability – as defined by Edgar Dale and Jeanne Chall (Dale and Chall, 1949) – is "the sum total (including all the interactions) of all those elements within a given piece of printed material that affect the success a group of readers has with it. The success is the extent to which they understand it, read it at an optimal speed, and find it interesting." Text readability has a huge impact on the reading and comprehending a text. Base on the readability, readers can determine whether a text is suitable for their reading ability or not. The text author(s) can also use the readability of the draft to guide readers object or have some adjustments to make it fit the toward reader.

Building a model to analyze text readability has meant a lot in the scientific and practical: help scientists writing research reports more readable; support educators drafting textbooks and curricula to suit each age of students; support publishers in shaping the audience; help governments drafting legal documents to suit the majority of citizens; or to assist manufacturers in preparing user guide for their products...In addition, text readability can effectively support in choosing appropriate curriculums when teaching language for foreigners.

Researches on text readability have begun since the early of the 20th century, most of them are for English and some common languages. Most famous studies in text readability are creating linear functions to assess and grade documents like Dale-Chall formula (Dale and Chall, 1949), Flesch Reading Ease formula (Flesch, 1949), Flesch-Kincaid formula (Kincaid et al., 1975), Gunning Fog formula (Robert, 1952), SMOG formula (Laughlin, 1969)...In Vietnamese, there are only two studies on text readability of Liem Thanh Nguyen and Alan B. Henkin in 1982 and 1985. Both these two researches focus on examining relations between statistical characteristics at words level and at sentences level and text readability. However, these two works only evaluate on small corpora of 20 documents (Nguyen and Henkin, 1982) and 54 documents (Nguyen and Henkin, 1985). Since these studies, there is almost no other publication on Vietnamese text readability.

In this paper, we mainly focus on creating a new formula for Vietnamese text readability assessment base on a self-built corpus with a large number of documents. Following this, the remaining of this paper is presented as below: we first present some famous formulas in English text readability as related works in section 2; we then present our works on building a corpus and creating formulas to assess Vietnamese text readability along with some experiments and results in section 3; finally, section 4 presents our discussions and conclusions.

## 2 Related works

Studies on text readability have begun since the early of the 20th century. Up to now, there are thousands of works in this field. In this section, we will describe some famous formulas for English text readability assessment and two formulas for Vietnamese.

First is the new Dale-Chall formula (Chall and Dale, 1995). This is a very famous readability formula that provides the comprehension difficulty score of an English document. This uses a list of 3000 words that students in fourth-grade could understand. Words that not in this list are considered as difficult words. The score is calculated as Equation 1:

$$RS = 0.1579 * (PDW) + 0.0496 * ASL \quad (1)$$

where $RS$ is the Reading Grade of a reader who can comprehend the text at 3rd grade or below, the higher $RS$, the more difficult the text; $PDW$ is Percentage of Difficult Words; and $ASL$ is Average Sentence Length in words. The list of 3.000 words that student in fourth-grade could understand was replaced by the list of 3.000 frequent Vietnamese words to experiment in Vietnamese corpus.

The second formula is Flesh Reading Ease (Flesch, 1949). This is an easy formula for measuring English text readability and is used by many US government agencies like US Department of Defense... It is also integrated into the Microsoft Word - the most popular word processor - from version 2007 to help users checking and controlling the readability of the document. The formula is defined as Equation 2:

$$RE = 206.835 - (1.015 * ASL) - (84.6 * ASW) \quad (2)$$

where $RE$ is the Readability Ease of the text; $ASL$ is Average Sentence Length in words; and $ASW$ is the average number of syllables per word. Like the name of the formula, the higher $RE$, the easier the document.

Next is the Flesch-Kincaid grade level readability formula (Kincaid et al., 1975) for English. This formula is best suited for education and is also integrated into the Microsoft Word. It is defined as Equation 3:

$$FKRA = (0.39 x ASL) + (11.8 * ASW) - 15.59 \quad (3)$$

where $FKRA$ is the Flesch-Kincaid Reading Age, indicates the grade-school level that student can read and comprehend; $ASL$ is Average Sentence Length in words; and $ASW$ is the average number of syllables per word. Because this formula determines the grade-school level of the text so the higher $FKRA$, the more difficult the text.

Continue is the Gunning Fog index formula (Robert, 1952). This formula is developed by Robert Gunning – an American textbook publisher. It is defined as Equation 4:

$$\text{Grade level} = 0.4 \left( ASL + PHW \right) \quad (4)$$

where $ASL$ is Average Sentence Length in words; and $PHW$ is Percentage of Difficult Words - the percentage of words which have three or more syllables that are not proper nouns, combinations of easy words or hyphenated words, or two-syllables verbs made into three with -es and -ed endings. Similar to the Flesch-Kincaid grade level, the higher Gunning Fog index, the more difficult the text.

Next is the SMOG formula (Laughlin, 1969). This formula estimates the years of education a person needs to understand a specific text and is widely used in checking health messages. The higher SMOG value, the more difficult the text. The formula is:

$$\text{SMOG grade} = 3 + \sqrt{\text{Polysyllable Count}} \quad (5)$$

In Vietnamese, as mentioned in Section 1, there are only two researches on text readability of Liem Thanh Nguyen and Alan B. Henkin in 1982 and 1985 (Nguyen and Henkin, 1982, 1985). The first formula is defined as Equation 6

$$RL = 2WL + 0.2SL - 6 \quad (6)$$

where $RL$ is Readability Level of the text; $WL$ is Average Word Length in characters; and $SL$ is Average Sentence Length in words. The second formula was revised from the first one with the additional role of the ratio of difficult words in the document:

$$RL = 0.27WD + 0.13SL + 1.74 \quad (7)$$

where $RL$ is Readability Level of the text, the higher $RL$ value, the more difficult the text; $WD$ is Word Difficulty; and $SL$ is Average Sentence Length in words.

However, these two works only evaluate on small corpora of 20 documents (Nguyen and Henkin, 1982) and 54 documents (Nguyen and Henkin, 1985). Since these studies, there is almost no other publication on Vietnamese text readability. Furthermore, as proposed by Professor Lucius Adelno Sherman, languages are still changing over time (Sherman, 1893), so researches on readability will still continue.

## 3 Method

### 3.1 Building corpus

As mentioned in sections 1 and 2, there are only two publications on Vietnamese text readability and both of them use only small corpora with 20 and 54 documents (Nguyen and Henkin, 1982, 1985). So in our research, we built another corpus with a larger amount of document for examining. 1,000 documents were collected into 3 categories of difficulty from various sources with the following criteria:

1. Easy documents: including documents written for children or by children or just need people who are studying at primary schools or having maximum primary education to read and understand. These documents were mainly collected from primary school textbooks, primary sample essays, fairy-tales, stories for babies...

2. Normal documents: they are documents written for middle and high school students, or documents which only need people with high school education to be readable and understandable. Most documents in this category were collected from textbooks and general newspapers.

3. Difficult documents: including documents written for college students, specialized documents, scientific paper...which need high or specialized education to be readable and understandable. These documents were collected from university textbooks, specialized documents, political theory articles, language and literary articles, law and legal documents...

Ten experts were asked to evaluate collected documents. They are Vietnamese language specialists, current or former Vietnamese literature teacher - who has much knowledge and experiment in using and teaching Vietnamese. Each document was ensured to be evaluated by 3 experts with the following instruction: each person carefully reads each given document and gives a score for that document: *a*) If that document is easy enough for children and people with just primary school education to read and understand, give the Score = 1; *b*) Score = 2 if almost all adults and people with average education can read and understand; *c*) If documents need people with high or specialized education to be readable and understandable, give the Score = 3. The overall scores of each document are used to re-category that document:

1. If the document got 3 equal scores: that document will be categorized according to the category of the score.

2. If the document got 2 equal scores and one different: that document will be put to the category of the 2 equal scores.

3. If the document got 3 different scores: that document will be filtered out of the corpus.

|  | Easy | Normal | Difficult | Overall |
|---|---|---|---|---|
| **No. documents** | 235 | 413 | 348 | 996 |
| **No. sentences** | 4,006 | 13,772 | 24,646 | 42,424 |
| **Average sentence length in word** | 13.59 | 19.24 | 22.71 | 19.12 |
| **Average sentence length in syllable** | 16.48 | 25.04 | 33.66 | 26.03 |
| **Average sentence length in character** | 79.50 | 124.39 | 173.54 | 130.97 |
| **Average word length in syllable** | 1.21 | 1.30 | 1.48 | 1.34 |
| **Average word length in character** | 5.82 | 6.44 | 7.62 | 6.71 |

Table 1: Statistics numbers of built corpus.

Finally, for some statistic later, we assign the readability value of 1 for all documents in the Easy category, 2 for the Normal and 3 for the Difficult category. Table 1 presents some statistic number of the final corpus.

To ensure that our corpus is reliable, we calculated the Fleiss' kappa score (Fleiss, 1971). This is a statistical measure for assessing the agreement between a fixed number of raters when classifying items. The Fleiss' kappa measure is usually used in natural language processing to assess the reliability of agreement between references in the corpus or between manual annotation by raters/experts. The K value is 0.422, which demonstrates that there is a moderate agreement between annotators so the corpus is reliable.

### 3.2 Features

In this part, we will describe some features that are commonly used in text readability assessment. **Average sentence length:** the average sentence length of a text is one of the simplest and common characteristic when measuring text readability. In this paper, we examined three types of average sentence length as Equation 8, 9 and 10:

Average Sentence Length in Words (**ASLW**):

$$ASLW = \frac{\text{word count}}{\text{sentence count}} \quad (8)$$

Average Sentence Length in Syllables (**ASLS**):

$$ASLS = \frac{\text{syllable count}}{\text{sentence count}} \quad (9)$$

Average Sentence Length in Characters (**ASLC**):

$$ASLC = \frac{\text{character count}}{\text{sentence count}} \quad (10)$$

**Average word length:** two types of average word length were examined in this study as Equation 11 and 12:

Average Word Length in Syllables (**AWLS**):

$$AWLS = \frac{\text{syllable count}}{\text{word count}} \quad (11)$$

Average Word Length in Characters (**AWLC**):

$$AWLC = \frac{\text{character count}}{\text{word count}} \quad (12)$$

In Vietnamese, word maybe monosyllabic or polysyllabic (compound word) with a whitespace between each syllable; each syllable is a combination of letters with or without tonal and word marks. For example: the word "*nghe*" (listen) is a monosyllable with four letters (*n*, *g*, *h*, and *e*); the word "*chạy*" (run) is a monosyllable with four letters (*c*, *h*, *a*, *y*) and a tonal mark (.); the word "*thời gian*" (time) is a polysyllable (2 syllables "*thời*" and "*gian*") with eight letters (*t*, *h*, *o*, *i*, *g*, *i*, *a*, *n*), a word mark ( ' ), a tonal mark ( ` ) and a whitespace.

**Percentage of difficult words:** in many studies, the percentage of difficult words is an important feature when evaluating text readability. However, create the easy or difficult word list needs a lot of effort, so most researches used frequent word list as a replacement: if a word does not appear in the frequent list, it will be considered as a difficult word. In this study, we used two frequent lists: the first is top 1,000 frequent words extracted from the frequent word list of Dien and Hao (Dinh and Hao, 2015); and the second is 1,000 frequent words extracted from all easy documents of our built corpus. The percentage of difficult words is calculated as Equation 13:

$$PDWi = \frac{\text{difficult word count}}{\text{word count}} \quad (13)$$

In this paper, **PDW1** stands for a percentage of difficult words calculated using Dien and Hao's list and **PDW2** stands for a percentage of difficult words calculated using our easy list.

Not only the percentage of difficult words, but also the **percentage of difficult syllables** was examined in our study. We also used two frequent lists: the top 1,000 frequent syllables extracted from the list of Dien and Hao (Dinh and Hao, 2015); and the 1,000 frequent syllables extracted from all easy documents of our corpus. The percentage of difficult syllables is calculated as Equation 14:

$$PDWi = \frac{\text{difficult syllable count}}{\text{syllable count}} \quad (14)$$

Similar to word, in this paper, **PDS1** and **PDS2** stand for a percentage of difficult words calculated using Dien and Hao's list and our easy list respectively.

### 3.3 Create formula

The first thing is finding which features are suitable for predicting text readability through corre-

|  | TR | ASLW | ASLS | ASLC | AWLS | AWLC | PDS1 | PDW1 | PDS2 | PDW2 |
|---|---|---|---|---|---|---|---|---|---|---|
| **TR** | 1 | | | | | | | | | |
| **ASLW** | 0.567 | 1 | | | | | | | | |
| **ASLS** | 0.675 | 0.970 | 1 | | | | | | | |
| **ASLC** | **0.695** | 0.955 | 0.997 | 1 | | | | | | |
| **AWLS** | 0.770 | 0.483 | 0.669 | 0.699 | 1 | | | | | |
| **AWLC** | **0.774** | 0.497 | 0.674 | 0.716 | 0.979 | 1 | | | | |
| **PDS1** | 0.051 | -0.116 | -0.140 | -0.132 | -0.107 | -0.073 | 1 | | | |
| **PDW1** | 0.103 | -0.069 | -0.029 | -0.018 | 0.156 | 0.156 | 0.680 | 1 | | |
| **PDS2** | 0.466 | 0.201 | 0.237 | 0.251 | 0.328 | 0.355 | 0.666 | 0.602 | 1 | |
| **PDW2** | **0.786** | 0.438 | 0.562 | 0.589 | 0.785 | 0.793 | 0.340 | 0.467 | 0.782 | 1 |

Table 2: Correlation between each feature with the text readability and between features together (TR is the Text Readability).

|  | Fold 1 | Fold 2 | Fold 3 | Fold 4 | Fold 5 | Average |
|---|---|---|---|---|---|---|
| | **Coefficient values** | | | | | |
| **Const** | -0.8141 | -0.6767 | -0.6372 | -0.7777 | -0.7419 | **-0.7295** |
| **ASLC** | 0.0037 | 0.0040 | 0.0043 | 0.0040 | 0.0040 | **0.0040** |
| **AWLC** | 0.2105 | 0.1779 | 0.1699 | 0.2032 | 0.1911 | **0.1905** |
| **PDW2** | 2.6960 | 2.7974 | 2.7224 | 2.6370 | 2.7208 | **2.7147** |
| | **Accuracy** | | | | | |
| **Easy** | 0.7234 | 0.7021 | 0.8511 | 0.7234 | 0.8085 | **0.7660** |
| **Normal** | 0.8313 | 0.9036 | 0.8537 | 0.8537 | 0.8313 | **0.8571** |
| **Difficult** | 0.7681 | 0.7857 | 0.8000 | 0.8000 | 0.7826 | **0.7845** |
| **Overall** | 0.7839 | 0.8150 | 0.8342 | 0.8040 | 0.8090 | **0.8102** |

Table 3: Coefficient values and accuracy of all formulas.

lation analysis. Table 2 shows the correlation coefficients between each feature with the text readability and between features together performed on our corpus.

We can see that the ASLS, ASLC, AWLS, AWLC and PDW2 are high correlated with the text readability. However, they are also high correlated with some others. To choose which features to put in the formula, we select sequentially features from the highest correlation with text readability to the lowest and remove features have high correlation coefficient with selected features. Through some experiments, we decided to use the threshold 0.9 for removing features: if a feature has the correlation coefficient with selected features greater than or equal to 0.9, that feature will not be cho-

sen. Following this, three features were selected: ASLC, AWLC, and PDW2. The features AWLS and ASLS are high correlated with AWLC and ASLC so they were not chosen.

The selected features were used as predictors to perform multiple regression analysis with Text Readability as the criterion. The purpose is to find coefficient values of these features to form a formula for predicting text readability. The general formula is:

$$Readability = A_1 ASLC + A_2 AWLC$$
$$+ A_3 PDW2 + A_4 \quad (15)$$

With $A_1$ to $A_4$ are coefficient values. In this study, we divided our corpus into five equal parts for cross-fold analyzing. In each fold, four part were

selected to perform regression analysis and the co-efficient values of each fold will be used to predict the readability of the remaining part using Formula 15. The predicted values were rounded to the nearest unit and compared to the expert evaluated readability for accuracy assessment. Finally, the average coefficient value of each predictor was used to form the final text readability formula. The final formula is:

$$Readability = 0.004ASLC + 0.1905AWLC$$
$$+ 2.7147PDW2 - 0.7295$$
$$(16)$$

Table 3 presents coefficient values and accuracy of all formulas.

## 4   Discussion and conclusion

From the Table 2, we can see that word length, sentence length and the percentage of difficult words still play an important role in measuring the text readability. The Percentage of difficult syllables (PDS1) and words (PDW1) calculated on the lists extracted from Dien and Hao's lists (Dinh and Hao, 2015) are low correlated with the Text readability of the built corpus. The main reason is the lists of Dien and Hao were statistically analyzed from the corpus mainly collected from newspapers, which are mostly texts with normal readability. So the Dien and Hao's lists and other frequent lists which were collected from normal texts may not be good replacements for the easy word and syllable list.

In the Table 3, we can see that the final text readability formula can accurately predict more than 76% of the easy texts, more than 85% of the normal documents and more than 78% of the difficult texts. Overall, the formula can predict the Vietnamese text readability with the accuracy up to 81%. This is a good result and can be applied in practice.

In this paper, we have presented our work on creating a new large corpus for Vietnamese text readability assessing. We also used the corpus to create a new formula for predicting Vietnamese text readability. Experiments performed on the corpus using created formula shows that the formula can predict the readability of Vietnamese text with high accuracy.

For the future works, other corpora will be built with more detailed levels of difficulty and for more specific domains. Other deeper features like part-of-speech, sentence structure, discourse... will be examined to create more precise formula(s). Some machine learning methods will be examined to create some classifier for automatically Vietnamese text readability assessment.

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
