# Peer review of "A New Formula for Vietnamese Text Readability Assessment"

_ACL 2017 — decision unknown_

[Official Review · Reviewer 1 · rating 2 · confidence 4]
soundness 5 · originality 3 · clarity 4 · impact 3 · substance 2 · appropriateness 4 · meaningful comparison 4 · presentation format Poster

- Strengths: The paper broadens the applicability of readability scores to an
additional language, and produces a well-validated applicability score for
Vietnamese. 

- Weaknesses: The greatest weaknesses, with respect to ACL are that 1)
readability scores are of limited interest within the field of computational
linguistics. While they are somewhat useful in educational and public
communication fields, their impact on the progress of computational linguistics
is limited.  A minor weakness is in the writing: the paper has numerous minor
grammatical errors.
Although the discussion compares the performance of the PDS1 and PDW1 features
from the previous work, it is unclear how poorly the previous readability
measures perform, relevant to the one developed here, for practical purposes.

- General Discussion: This paper would be a stronger candidate for inclusion if
the corpus (and importantly, labels developed) were released. It could be used
more widely than the development of scalar readability metrics, and would
enable (e.g.) investigation of application of more powerful feature-selection
methods.

[Official Review · Reviewer 2 · rating 1 · confidence 5]
soundness 5 · originality 3 · clarity 4 · impact 3 · substance 1 · appropriateness 4 · meaningful comparison 4 · presentation format Poster

- Strengths:
 New Dataset, 
 NLP on Resource poor language

- Weaknesses:
 Incomplete related work references, 
 No comparison with recent methods and approaches, 
 Lack of technical contribution, 
 Weak experiments,

- General Discussion:

In this paper the authors present a simple formula for readability assessment
of Vietnamese Text. Using a combination of features such as word count,
sentence length etc they train a simple regression model to estimate the
readability of the documents. 

One of the major weaknesses of the paper its lack of technical contribution -
while early work in readability assessment employed simple methods like the one
outlined in this paper, recent work on predicting readability uses more robust
methods that rely on language models for instance (Eg :
http://www.cl.cam.ac.uk/~mx223/readability_bea_2016.pdf,
http://www-personal.umich.edu/~kevynct/pubs/ITL-readability-invited-article-v10
-camera.pdf). A comparison with such methods could be a useful contribution and
make the paper stronger especially if simple methods such as those outlined in
this paper can compete with more complicated models. 

Baseline experiments with SMOG, Gunning Fog index etc should also be presented
as well as the other Vietnamese metrics and datasets that the authors cite. 

Another problem is that while previous readability indices were more selective
and classified content into granular levels corresponding to grade levels (for
instance), the authors use a coarse classification scheme to label documents as
easy, medium and hard which makes the metric uninteresting. (Also, why not use
a classifier?)

The work is probably a bit too pre-mature and suffers from significant
weaknesses to be accepted at this stage. I would encourage the authors to
incorporate suggested feedback to make it better. 

The paper also has quite a few grammatical errors which should be addressed in
any future submission.

[Official Review · Reviewer 3 · rating 1 · confidence 4]
soundness 5 · originality 3 · clarity 3 · impact 3 · substance 1 · appropriateness 3 · meaningful comparison 4 · presentation format Poster

The authors present a new formula for assessing readability of Vietnamese
texts. The formula is developed based on a multiple regression analysis with
three features. Furthermore, the authors have developed and annotated a new
text corpus with three readability classes (easy, middle, hard).

Research on languages other than English is interesting and important,
especially when it comes to low-resource languages. Therefore, the corpus might
be a nice additional resource for research (but it seems that the authors will
not publish it - is that right?). However, I don't think the paper is
convincing in its current shape or will influence future research. Here are my
reasons:

- The authors provide no reasons why there is a need for delevoping a new
formula for readability assessments, given that there already exist two
formulas for Vietnamese with almost the same features. What are the
disadvantages of those formulas and why is the new formula presented in this
paper better?

- In general, the experimental section lacks comparisons with previous work and
analysis of results. The authors claim that the accuracy of their formula (81%
on their corpus) is "good and can be applied in practice". What would be the
accuracy of other formulas that already exist and what are the pros and cons of
those existing formulas compared to the new one?

- As mentioned before, an analysis of results is missing, e.g. which word /
sentence lengths / number of difficult words are considered as easy/middle/hard
by their model?

- A few examples how their formula could be applied in a practical application
would be nice as well.

- The related work section is rather a "background" section since it only
presents previously published formulas. What I'm missing is a more general
discussion of related work. There are some papers that might be interesting for
that, e.g., DuBay 2004: "The principles of readability", or Rabin 1988:
"Determining difficulty levels of text written in languages other than English"

- Since Vietnamese is syllable-based and not word-based, I'm wondering how the
authors get "words" in their study. Do they use a particular approach for
merging syllables? And if yes, which approach do they use and what's the
accuracy of the approach?

- All in all, the content of the paper (experiments, comparisons, analysis,
discussion, related work) is not enough for a long paper.

Additional remarks:

- The language needs improvements

- Equations: The usage of parentheses and multiplying operators is inconsistent

- Related works section: The usage of capitalized first letters is inconsistent